# Static Residual Tensile Strength Response of GFRP Composite Laminates Subjected to Low-Velocity Impact

**Jong-Il Kim [1], Yong-Hak Huh [1,\*] and Yong-Hwan Kim [2,\*]**

[1] Interdisciplinary Materials Measurement Institute (IMMI), Korea Research Institute of Standards and Science, Daejeon 34113, Korea; jikim@kriss.re.kr

[2] School of Mechanical Engineering, Chungnam National University, Daejeon 34134, Korea

\* Correspondence: yhhuh@kriss.re.kr (Y.-H.H.); yonghkim@cnu.ac.kr (Y.-H.K.)

**Abstract:** The dependency of the static residual tensile strength for the Glass Fiber-Reinforced Plastic (GFRP) laminates after impact on the impact energy level and indent shape is investigated. In this study, two different laminates, unidirectional, $[0°_2]_s$) and TRI (tri-axial, $(\pm45°/0°)_2]_s$), were prepared using the vacuum infusion method, and an impact indent on the respective laminates was created at different energy levels with pyramidal and hemispherical impactors. Impact damage patterns, such as matrix cracking, delamination, debonding and fiber breakage, could be observed on the GFRP laminates by a scanning electron microscope (SEM), and it is found that those were dependent on the impactor head shape and laminate structure. Residual in-plane tensile strength of the impacted laminates was measured and the reduction of the strength is found to be dependent upon the impact damage patterns. Furthermore, in this study, stress concentrations in the vicinity of the indents were determined from full-field stress distribution obtained by three-dimensional Digital Image Correlation (3D DIC) measurement. It was found that the stress concentration was associated with the reduction of the residual strength for the GFRP laminates.

**Keywords:** GFRP (glass fiber-reinforced plastics) composite; impact damage; DIC (digital image correlation); stress field; residual strength

## 1. Introduction

It has been revealed that composite structures are considerably sensitive to damage caused by accidental impact events, especially those of a low velocity, during fabrication and in service [1], and the damage may not be easy to inspect visually or with Non-Destructive Evaluation (NDE) use. Therefore, in order to apply the composite laminates in the structures reliably and cost-effectively, it may be important to understand the impact response of the laminates.

The complex nature and anisotropic properties of the polymeric composites can lead to the formation of various types of damage, which may be associated with failure in complex patterns. Under the low-velocity impact loading, damage patterns, including delamination, matrix cracking, debonding, and fiber breakage, which are similar to common damage modes in composite laminates, occur, and the damage is significantly dependent on the impact energy levels and impactor shapes [2–6]. The consequences of the damage may greatly reduce the stiffness and residual strength of the laminates.

Mechanical performance of the impacted composite laminates could be characterized through inspecting the damage by nondestructive techniques such as ultrasonic C-scan, X-radiography and infrared thermography. The threshold of barely visible impact damage in the laminates was investigated with the ND techniques [7], and Tuo et al. [8,9] assessed the low-velocity impact damage induced at four

different impact energy levels with ultrasonic C-scan and three-dimensional Digital Image Correlation (3D DIC) system, and analyzed the damage evolution and failure mechanism with experimental and numerical approaches. Impact damage strength degradation and progress of the damage were also investigated. Prato et al. [10] evaluated the residual pseudo-ductile behavior in tension for carbon fiber laminates, and reported that the damage area increased progressively as the impact and absorbed energy increased, and the full-field longitudinal strain distribution was measured by the DIC (Digital Image Correlation) system. Husman et al. [11] analyzed the local impact resistance of the composite laminate with fracture mechanics model as a laminate containing a hole of the same diameter as the impacting diameter. It was also reported that the residual strength could be modelled as a function of indentation depth [12] and was dependent on impact energy for the carbon fiber composites [12,13]. Mitrevski et al. [5] investigated the effect of impactor shape on the resulting damage and residual strength for thin woven carbon/epoxy laminates, and it was found that the blunter impactor caused the large damage area, and the residual tensile strength for the impacted specimens was significantly reduced. The dependency of residual strength on the impact energy was investigated by modelling the strength as a function of impact energy [12], and it was revealed that the strength of Carbon Fiber-Reinforced Polymer (CFRP) laminate decreased as the impact energy increased [13]. Even for the plain weave Glass Fiber-Reinforced Polymer (GFRP) composite laminates, Kounain et al. [14] showed that the residual tensile strength decreased with the increase in the impact energy due to the increase of the damage region, and tensile properties like strain at failure and elastic modulus also had the same dependency. Compression after impact (CAI) strength for the composite laminates also was showed to have the same dependency on the impact energy [15].

In this study, the effect of impact conditions, including impact energy and indent shape, and layer structure on the impact response for the GFRP laminates after impact was investigated with measurement of stress concentration around the indents. In order to examine the dependency of the residual strength for the impacted GFRP laminates on the impact conditions, an impact indent was introduced on the two different layer-structured GFRP laminates at different energy levels with pyramidal and hemispherical impactors. From the in-plane tensile test of the impacted laminates, the reduction of the residual strength was measured. The impact damage patterns of the impact indents also were observed with a scanning electron microscope (SEM), and the impact damage patterns were examined. Moreover, full-field stress fields around the respective indents were determined with DIC measurement and the association of the stress concentration around the indents with the reduction of failure strength of the laminates was investigated.

## 2. Experimental Programme

### 2.1. Specimen and Introduction of Impact Indent

In this study, UD (unidirectional, $[0°_2]_s$) and TRI (tri-axial, $(\pm45°/0°)_2]_s$) Glass Fiber-Reinforced Plastic (GFRP) laminates were prepared. The laminates were fabricated by E-glass-fiber fabric (Ahlstrom Glassfiber Oy, Mikkeli, Fl) and epoxy resin system (EPIKOTE RIMR135 epoxy resin and RIMH137 hardener, Hexion, Ohio, US) with a thickness of 3.5 mm using vacuum infusion method. Here, the vacuum intensity was controlled at less than 30 mbar. After the infusion was completed, the laminate panels were left at room temperature for 12 h and then were post-cured at 80 °C for 8 h.

Tensile specimen was prepared according to ASTM D5083 [16] and ISO 527-4 [17], where the specimen dimension was $250 \times 25 \times 3.5$ mm$^3$, as shown in Figure 1a, and 0° fiber orientation of the specimen aligned longitudinally. In order to prevent grip failure, double end tabs 50 mm long were adhesively bonded on both gripping ends of the specimen.

A low-velocity impact indent was introduced into the composite laminates by using a drop-weight impact machine (DD Machine, KR) with full impact capacity of 110 J. Here, the machine had a crosshead mass of 7.8 kg and maximum drop height of 1.1 m. To prevent multiple impacts on the specimen by the impactor, a crosshead brake was installed on the test rig, which was operated by sensing

rebound of the impactor with optical sensors. In this study, hemispherical and pyramidal impactors were used to introduce different impact indent shapes into the specimens. The head radius of the hemispherical impactor was 6.35 mm and opposite face angle of the pyramidal impactor was 136° and dimension of the respective impactor are shown in Figure 1b,c. The impactor nose was aligned to the center of the tensile specimen, represented in Figure 1a, to locate the indentation at the center of the specimen. The specimen was fixed with two 10 mm wide clamping blocks. Here, each block covered the whole width of the specimen at the location 30 mm apart from the specimen center in the longitudinal direction. In this study, to produce different impact damage into the laminates, three different incident impact energy levels, including 4, 6 and 8 J, were used.

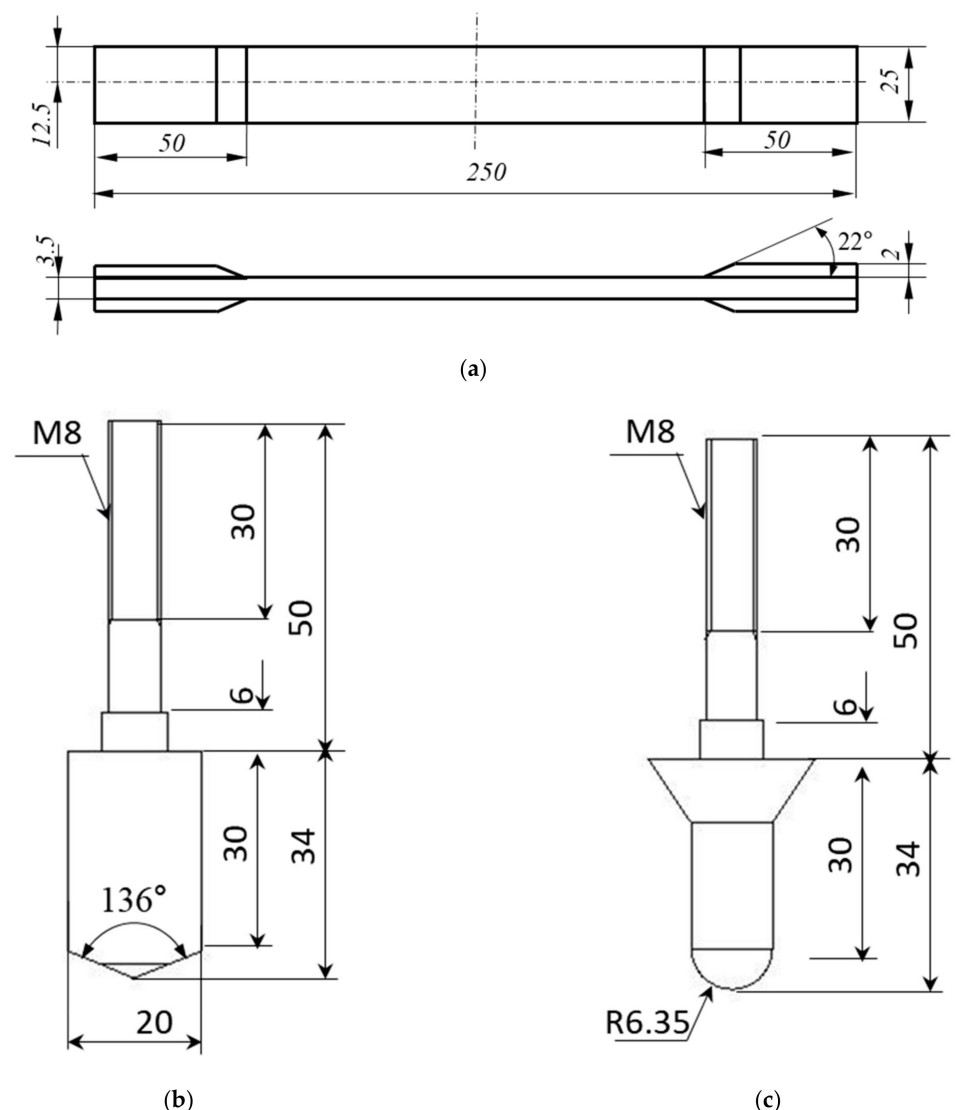

**Figure 1.** Geometry and dimension of (**a**) tensile specimen, (**b**) pyramidal impactor and (**c**) hemispherical impactor.

## 2.2. Test Method and DIC Measurement

In this study, in order to measure static in-plane tensile residual strengths of the GFRP laminates after impact, tension after impact tests of the impacted tensile specimens were carried out using a servo-hydraulic machine (MTS 810, MTS Systems, MN, USA) with a 100-kN load capacity. Figure 1a represents the geometry of the specimen. Tensile loads were imposed on the specimen in displacement control at a rate of 1 mm/min. by clamping both end tabs of the specimen with hydraulic wedge grips.

Before performing the tensile tests, load train alignment was adjusted with the maximum bending strain of less than 50 $\mu\varepsilon$. During tensile loading, the full-field in-plane strain distribution around the impact indent was measured using DIC (Digital Image Correlation) technique. In addition, tensile properties of the non-impacted UD and TRI GFRP laminates were measured according to ASTM D3039M and ISO 527-4, and those are presented in Table 1. Here, the tensile strain was obtained from the strain gages attached on the front and back face of the specimen. These tensile strength values were used in determining reduction in the residual strength due to impact damage

**Table 1.** Tensile properties of the unidirectional (UD) and tri-axial (TRI) Glass Fiber-Reinforced Plastic (GFRP) composite laminates.

| Laminate | Tensile Strength, $\sigma_{TS}$ (MPa) | | Elastic Modulus, E (GPa) | | Poisson's Ratio, $\nu$ | |
|---|---|---|---|---|---|---|
| | Longitudinal Direction | Transverse Direction | $E_{11}$ | $E_{22}$ | $\nu_{12}$ | $\nu_{21}$ |
| UD | $903.4 \pm 40.47$ | $39.7 \pm 2.25$ | 41.7 | 15.4 | 0.27 | 0.11 |
| TRI | $611.6 \pm 19.01$ | $133.7 \pm 2.85$ | 28.6 | 15.6 | 0.46 | 0.21 |

The DIC measurement was carried out with a commercial 3-dimensional DIC system (GOM, ARAMIS 5M) which was composed of two Charge Coupled Device (CCD) cameras, equipped with 35 mm Schneider lenses, with a resolution of $2448 \times 2050$. In this study, the angle between these two cameras was setup as $25°$, and the deformed surface images of the impacted laminate specimen were captured at a rate of 1 frame/s. The camera was calibrated with a 2.0 mm spaced black dot grid, which provided a resolution of 14.1 $\mu$m/pixel. Figure 2 shows the DIC system configuration for this measurement. In this DIC measurement, high contrast speckle pattern of white background and black dots was directly applied on the impacted surface of the specimen. The image was analyzed with the facet size and step size of 24 and 19 pixels, respectively. Here, the spatial resolution was 0.272 mm.

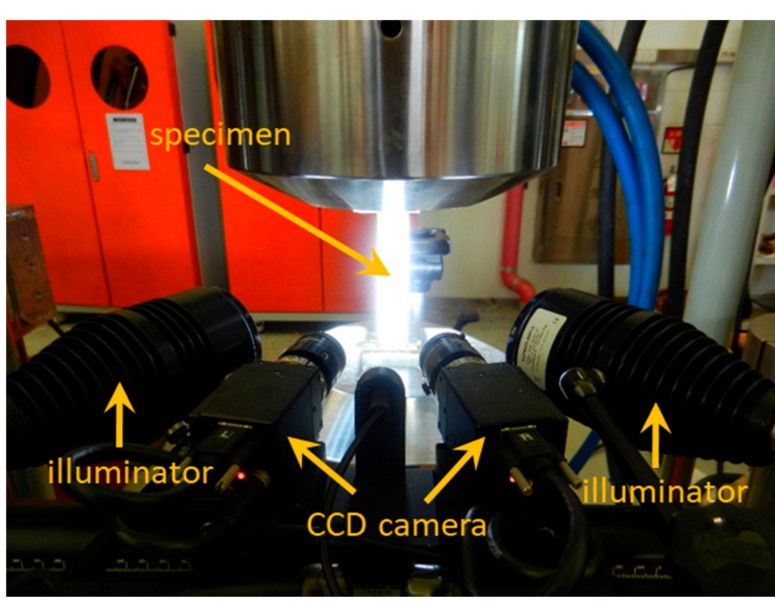

**Figure 2.** Experimental system configuration for digital image correlation (DIC) measurement.

### 2.3. SEM Observation

Impact damage created on the TRI and UD GFRP laminates was observed by a Field Emission Scanning Electronic Microscope (FE SEM, JEOL JSM-7100F, JP). In order to observe the damage pattern in three directions, the sample block, with a front impacted-surface, longitudinal and transversal cross-sections intersected at the indent center, was prepared by carefully cutting the longitudinal and transverse faces of the impacted laminates with diamond blade. Here, the cutting velocity and rotation

speed of the diamond blade were 1.27 mm/min and 1000 rpm, respectively. These two cuts were ground and mechanically polished from P1000 paper down to diamond compound of 1 μm. In this study, typical damage patterns produced by the respective hemispherical and pyramidal impactors for the UD and TRI laminates were observed. The SEM images on the respective surfaces were obtained at accelerating voltage of 1.0 kV under the vacuum pressure of $6.5 \times 10^{-4}$ Pa.

## 3. Results and Discussion

### 3.1. Impact Indent Geometry

In this study, two different indent shapes were introduced in the respective TRI and UD GFRP laminates using pyramidal and hemispherical head-shaped impactors at three different impact energy levels of 4, 6 and 8 J. Figure 3a,b show typical indent patterns from the pyramidal and hemispherical head impactors, respectively, on the impacted surface for both laminates. Cross-section height profiles of the respective indentations were measured using a surface roughness tester (Surftest, Mitutoyo), as shown in Figure 3c,d. Here, the profile was measured repeatedly three times. Profiles of the respective indentations had the same appearances as the corresponding impactor head shapes, irrespective of the laminate structures, and the typical pyramidal and hemispherical indent profiles are presented in Figure 3c,d. The dimensions of the indents were varied with the impact energy levels. However, for the pyramidal indent, the profile near the indent tip was partly flat, as shown in Figure 3c, which may result from the presence of fiber breaks which would be caused by a locally dynamic impact process. Fiber fracture damage in the composite laminate under impact load was revealed to be mainly affected by high local stress concentration near the impact point [18,19]. Impact load in the composite laminate can produce high local stress concentration near the impact point and indentation, and the energy can be dissipated through a damage process consisting of fiber fracture beneath the impact point and delamination in the locally deformed area. It was reported that the fiber breaks can increase with increasing velocity, depending on the severity of local deformation and fragmentation [18,19]. From these profile measurements, dependency of the impact indent geometry on the impactor shape, impact energy level and laminate direction could be investigated. In this study, the maximum depth, $z_{max}$, and width, $w_{max}$, of the respective indents were determined, which are plotted in Figure 4 and listed in Table 2. Here, the maximum depth and width are the maximum depth of the indent from the impacted surface and a square side length or diameter of the indent base at the impacted surface of the specimen, respectively. As found in Figure 4, the maximum depth, $z_{max}$, of the pyramidal indent for the TRI laminate was not dependent on the impact energy, while for the UD laminate, it was not varied at the relatively low impact energy level, but increased slightly with increasing impact energy at the relative high impact energy, 8 J. The depth, $z_{max}$, for the TRI and UD GFRP laminates, was within a range of between 111 to 117 μm, and between 164 to 207 μm, respectively. For the hemispherical indent, the depth, $z_{max}$, was nearly linearly dependent on the impact energy with a rate of 8 μm/J within a range of 46–77 μm and 10 μm/J within a range of 63–102 μm for the TRI and UD laminates, respectively. Comparing the depth for both indents, $z_{max}$ of the pyramidal indent was relatively larger than that of the hemispherical indent for both laminates and all impact energy levels. $z_{max}$ of the hemispherical indent ranged from 46 to 102 μm for both laminates, while $z_{max}$ for the pyramidal indent from 111 to 207 μm. Therefore, it can be found that the depth of the pyramidal indent is more than twice as deep as the hemispherical indent depth.

The width, $w_{max}$, of both indent shapes for the TRI laminate was within a range of between 4.39 and 5.94 mm and nearly linearly dependent on the impact energy level with a rate of 0.36 mm/J. For the UD laminate, $w_{max}$ of both indent shapes also increased with increasing impact energy level, but $w_{max}$ of the pyramidal indent became larger than that of the hemispherical indent with increasing impact energy. The linear rate of the width for the pyramidal and hemispherical indents was calculated as 0.65 and 0.2 mm/J, respectively. The width, $w_{max}$, for UD laminates was within a range of between 3.82 and 6.45 mm.

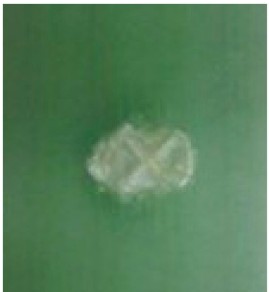

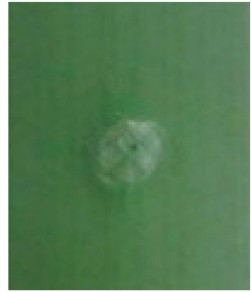

(**a**) Shape of the pyramidal impact indent          (**b**) Shape of the hemispherical impact indent

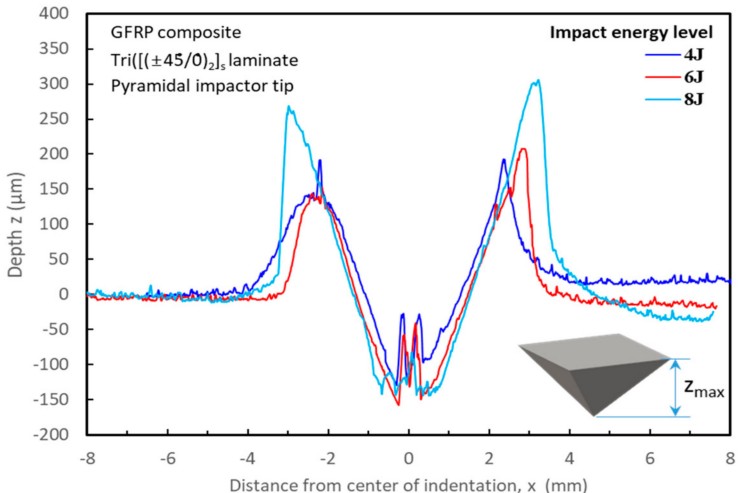

(**c**) Profile of the pyramidal impact indent

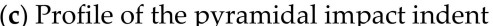

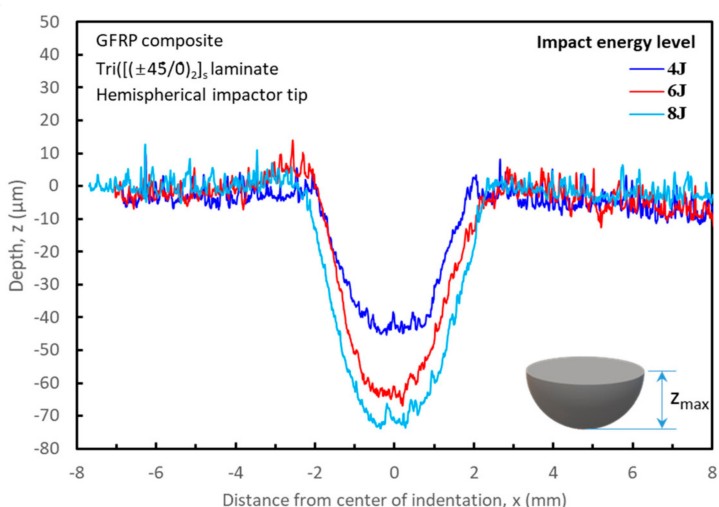

(**d**) Profile of the hemispherical impact indent

**Figure 3.** Typical shape and profile of the pyramidal and hemispherical impact indents.

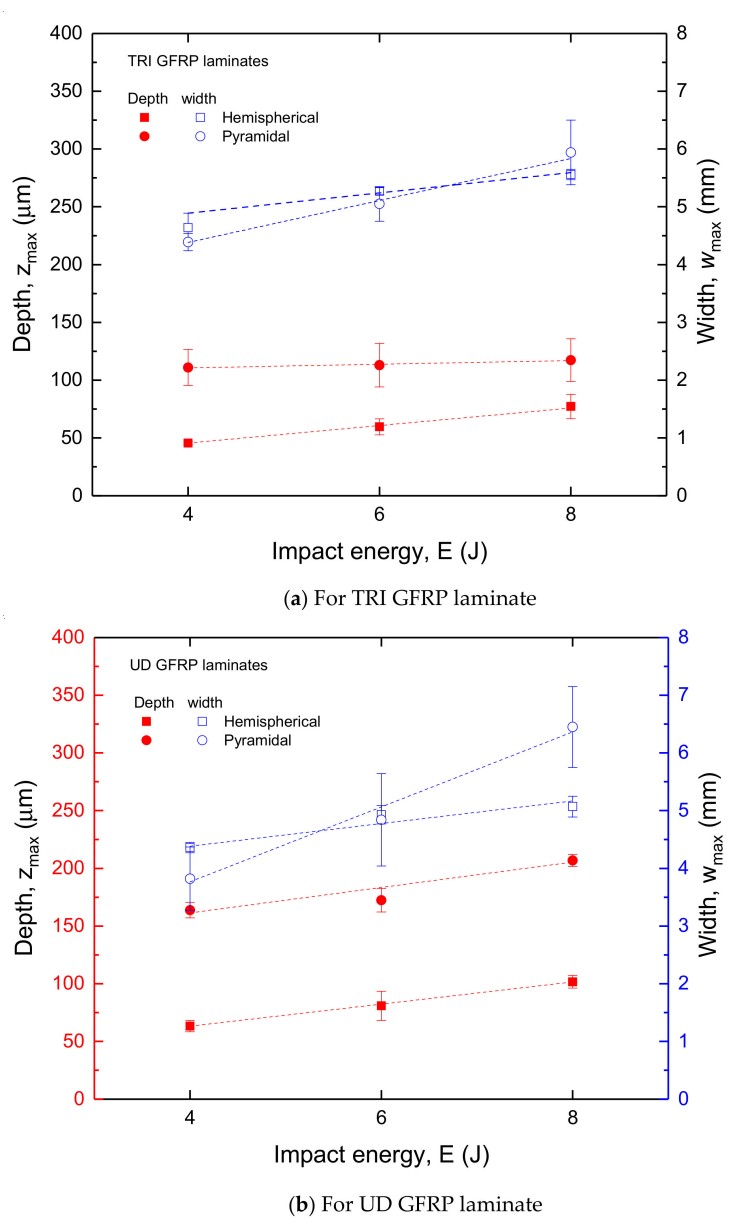

(**a**) For TRI GFRP laminate

(**b**) For UD GFRP laminate

**Figure 4.** Maximum depth and width of the pyramidal and hemispherical indent (**a**) for the TRI (**b**) UD GFRP laminates.

**Table 2.** Dimension of impact indent profiles created in the UD and TRI laminates.

| | Laminate Type | TRI | | UD | |
|---|---|---|---|---|---|
| Impact Energy (J) | Head Shape | Hemispherical | Pyramidal | Hemispherical | Pyramidal |
| Width, $w_{max}$ (mm) | 4 | 4.64 ± 0.25 | 4.39 ± 0.15 | 4.36 ± 0.09 | 3.82 ± 0.55 |
| | 6 | 5.27 ± 0.06 | 5.05 ± 0.3 | 4.93 ± 0.16 | 4.84 ± 0.8 |
| | 8 | 5.56 ± 0.09 | 5.94 ± 0.56 | 5.07 ± 0.18 | 6.45 ± 0.7 |
| Depth, $z_{max}$ (μm) | 4 | 45.6 ± 1.6 | 110.9 ± 15.5 | 63.3 ± 4.9 | 163.8 ± 6.7 |
| | 6 | 59.6 ± 6.9 | 113 ± 18.9 | 80.9 ± 12.6 | 172.4 ± 10.2 |
| | 8 | 77.2 ± 10.6 | 117.3 ± 18.5 | 101.7 ± 5.5 | 206.8 ± 5.2 |

### 3.2. Observation of Low-Velocity Impact Damage Pattern

As shown in Figures 3 and 4, the impact indent geometry was dependent on the impactor head shape, laminate direction and impact energy level. The impact damage mode generated below the impact indent under the impact may be another significant consideration for the GFRP composite design. In general, major impact damage modes have been reported to include matrix cracking, delamination and fiber breakage [8,9].

In this study, the damage pattern for the TRI and UD GFRP laminates impacted with different head-shaped impactors at different energy levels was examined. Figures 5 and 6 represent typical impact damage patterns generated beneath the indents in TRI and UD laminates. The patterns were observed on the sectional planes along the longitudinal and transvers direction of the laminates with FE SEM (Field Emission Scanning Electronic Microscope, JEOL). Figure 5 represents damage patterns created from the pyramidal impactor for the TRI GFRP laminate, where those were incurred at the impact energy level of 8 J. Here, a 3D sectional view, assembling the damage aspects on the front, longitudinal and transverse plane, and three magnified views A, B and C, representing the damage patterns produced at the local areas A, B and C on these three directional planes, respectively, are displayed. On the impacted front plane, as can be seen in the magnified view A, fiber breakage and matrix splitting were found. These modes may be induced by stress concentration at the contact of the impactor edges. On the longitudinal and transverse planes, various damage modes, including matrix cracking, fiber breakage, debonding and delamination, were generated, as shown in the magnified views B and C in Figure 5. In this study, the damage patterns from the pyramidal impactor in UD GFRP laminate also were observed, and it was found that the dominant patterns were considerably identical to the damage patterns observed in TRI laminate. Typical damage patterns from the hemispherical impactor are presented in Figure 6. In Figure 6, the damage patterns created in the UD GFRP laminate at impact energy of 8 J are displayed. Here, like Figure 5, a 3D sectional view assembling the damage aspects along the three directions and three magnified views A, B and C on these three directional planes, respectively, are represented. Contrary to the damage from the pyramidal impactor, the damage from the hemispherical impactor was not created just beneath the impact indent, as can be found in Figure 6. Matrix cracking and debonding between matrix and fiber were found below the indent, as shown in the magnified views B and C of Figure 6.

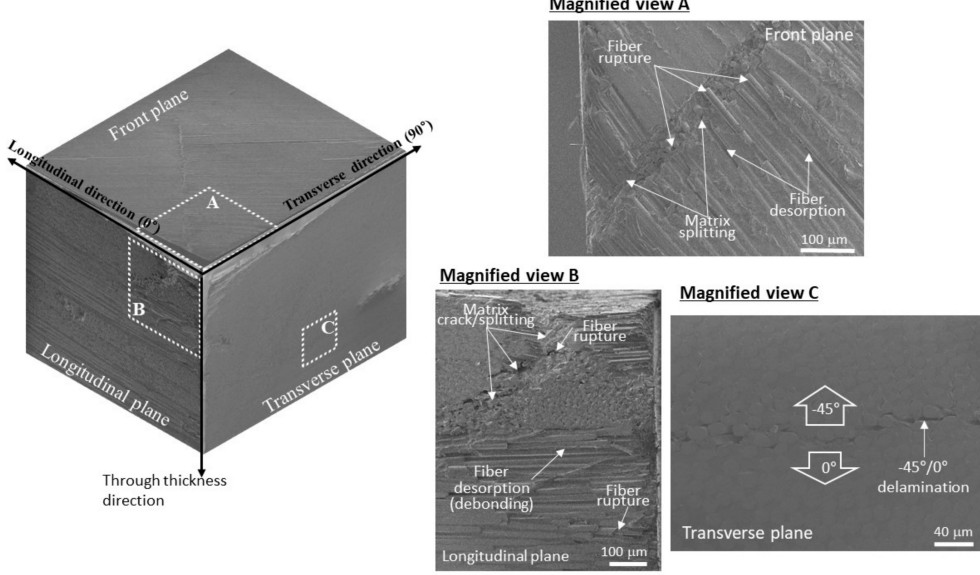

**Figure 5.** Typical impact damage patterns created by a pyramidal impactor for TRI laminate (at impact energy level of 8 J).

Based on the SEM observation presented in Figures 5 and 6, dominant impact damage patterns, which can be created in the GFRP laminates, can be schematically represented as shown in Figure 7. The dominant impact damage modes are dependent on the impactor head shape and laminate structure, but it can be said that the damage patterns, such as matrix cracking, delamination, debonding and fiber breakage, predominantly occurred in GFRP laminates.

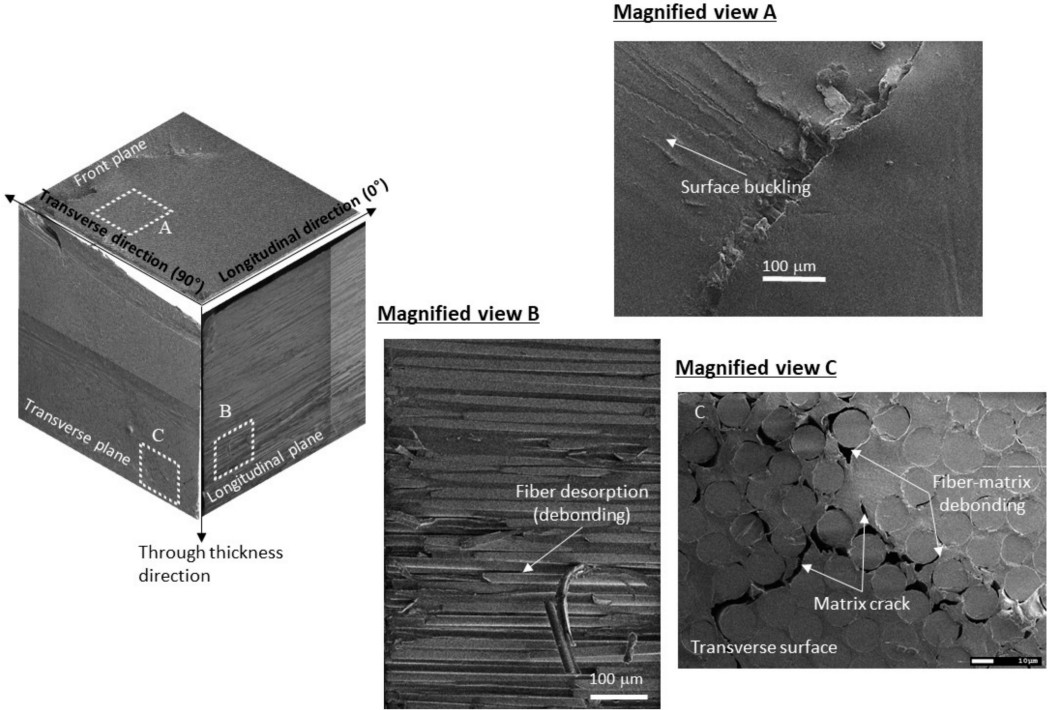

**Figure 6.** Typical impact damage patterns created by a hemispherical impactor for UD laminate (at impact energy level of 8 J).

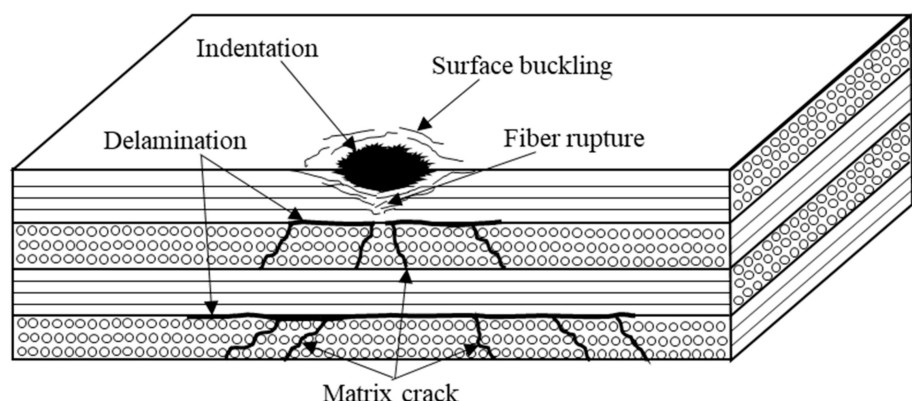

**Figure 7.** Schematic representation of the dominant impact damage patterns created in TRI and UD GFRP laminates.

### 3.3. Reduction of Residual Static Strength

As shown in Figure 4, Figure 5, Figure 6, the low-velocity impact created various forms of internal damage patterns, including matrix crack, fiber breakage, delamination, and debonding, and indents on the impacted surface of the GFRP laminates. These indents and damage can cause significant strength reduction, which also can lead to unexpected failure of the laminates [13].

In this study, the dependency of the residual strength of the impacted GFRP laminates on the laminate structure, impact energy, and indent shape was examined, which is represented in Figure 8. Here, the residual in-plane tensile strength of the impacted TRI and UD laminates was determined as a failure strength obtained from in-plane tensile test of the impacted laminates, and obtained from duplicate tests. In Figure 8, the strengths were normalized with the tensile strength of the respective laminates, which was displayed as the residual strength ratio, $\sigma_f/\sigma_{TS}$. Table 3 lists the numeric values of the ratio. As can be found from Figure 8, the residual strength for the GFRP laminates was reduced with increasing impact energy. For the TRI GFRP laminates impacted with hemispherical and pyramidal impactors, the ratios were linearly decreased with increasing impact energy, irrespective of impactor head shape, where the decreasing rates were 0.0236/J and 0.0126/J, respectively. The reduction of the ratios, ranging from 0.87 to 0.82, for the pyramidal indentation was considerably lower than that, ranging from 0.97 to 0.82, for the hemispherical indent. Therefore, it can be said that the residual strength for the TRI GFRP laminate may be considerably more sensitive to the pyramidal impact indent, compared to the hemispherical impact indent. This strength reduction with increasing impact energy is likely to be closely related to the indentation dimension increment with increasing impact energy, which is shown in Figure 4. The indentation depth and width on the impacted surface were created larger with increasing impact energy, as shown in Figure 4. It had been reported that the impact residual strength decreased as the impact energy and impact indent dimension for CFRP laminates increased [13].

For the UD GFRP laminate impacted with the hemispherical impactor, the residual strength was reduced by 3–4%, and was not considerably changed with increasing impact energy. Contrary to the hemispherical indent, the residual strength of the laminate impacted with pyramidal impactor was relatively greatly reduced. Especially, the strength reduction ratio for the UD laminate impacted at less than 6 J was nearly similar to that for the hemispherical indent, but, at impact energy of 8 J, the ratio dropped down to 86 %. This change of the strength ratio may also be closely related to the dramatic change in pyramidal indentation dimension at greater than impact energy of 6 J. As can be found in Figure 4, the pyramidal indentation dimension was dramatically increased over the impact energy range of more than 6 J.

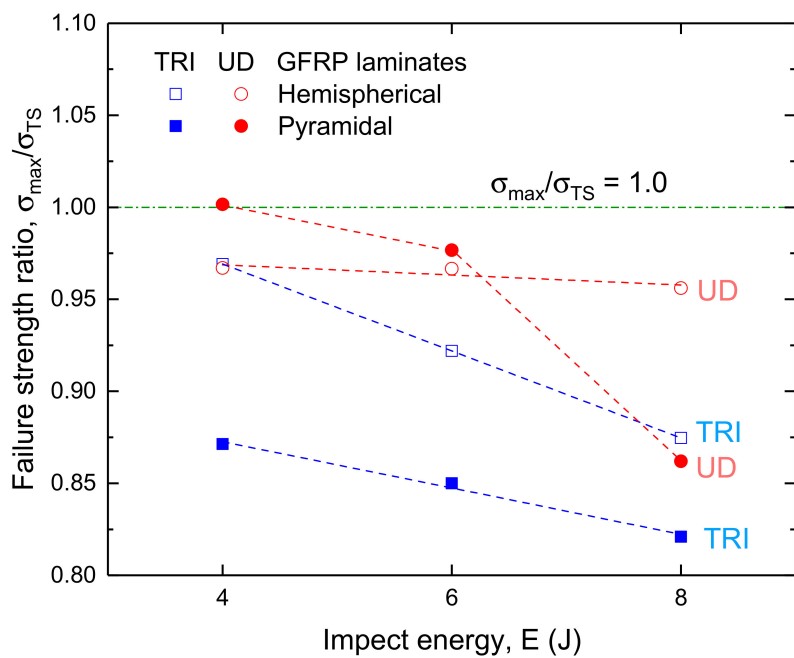

**Figure 8.** Dependency of residual strength of the impacted GFRP laminates on the impact energy and impact indent shape.

**Table 3.** Residual tensile strength reduction ratio for the impacted UD and TRI GFRP laminates.

| Laminate Type | TRI | | UD | |
|---|---|---|---|---|
| Head Shape<br>Impact Energy (J) | Hemispherical | Pyramidal | Hemispherical | Pyramidal |
| 4 | 0.97 | 0.87 | 0.97 | 1.00 |
| 6 | 0.92 | 0.85 | 0.97 | 0.98 |
| 8 | 0.87 | 0.82 | 0.96 | 0.86 |

Therefore, it is found that the residual strength of the TRI GFRP laminate is much more sensitive to the impact damage, compared to the UD laminate, and the pyramidal impact indent has greater influence on the strength than the hemispherical impact indent.

*3.4. Stress Concentration Around the Impact Indent*

As can be found from Figure 8, the residual strengths of the impacted GFRP laminates were dependent on the impact conditions such as impact energy and impactor head shape. The impact conditions created various impact damage forms and indent geometries, as shown in Figure 4, which may influence on the failure strength with a highly localized non-uniform stress distribution around the indent. Therefore, information on the stress distribution around the impact intent may be considerably significant to understand a failure response of the impacted laminate. As a result, in this study, local stress distribution around the impact indent on the impacted laminate surface was determined under in-plane tensile loading using DIC technique. Prato et al. [10] also had measured longitudinal surface strain map on the impacted carbon fiber laminates with DIC technique and locally concentrated strain around the hemispherical indent was measured.

Figures 9 and 10 represent typical stress distribution around the impact indent for TRI and UD GFRP laminates, respectively. Here, the stress distribution was determined from the DIC measurement using linear elastic orthotropic relationship. In Figure 9, the directional stress distributions around the pyramidal and hemispherical indent for the TRI GFRP laminate are presented. Here, the full-field stress distributions were obtained at the in-plane tensile stress of 600 MPa which was equivalent to 0.82 $\sigma_{TS}$ (tensile strength of the TRI GFRP laminate). As shown in Figure 9, the longitudinal ($\sigma_{yy}$), transverse ($\sigma_{xx}$) and shear ($\sigma_{xy}$) stresses around the pyramidal indent were concentrated in the vicinity of the indent and the concentrated stress became higher with increasing impact energy. The concentration zone envelope of the longitudinal stress, which was developed with an angle of ±45°, was expanded larger with increasing impact energy. Transverse and shear stresses were concentrated in the vicinity of the indent, and the concentration zones of the respective stresses also were expanded with increasing impact energy. In comparison, for the hemispherical indentation, all directional stresses were not remarkably concentrated, as can be found in Figure 9. Longitudinal, transverse and shear stresses were not relatively highly concentrated in the vicinity of the indent.

Figure 10 represents the directional stress distributions around the pyramidal and hemispherical indent for the UD GFRP laminate, which were obtained at the tensile stress of 700 MPa (equivalent to 0.78 $\sigma_{TS}$, where $\sigma_{TS}$ represents tensile strength of the UD laminate). As shown in Figure 10, the longitudinal ($\sigma_{yy}$) and transverse ($\sigma_{xx}$) stresses around the pyramidal indentation were not relatively remarkably concentrated at the indent edge, unlike those for the TRI laminate, but shear ($\sigma_{xy}$) stress concentration zone was developed, similar to the TRI laminate. Similar to the pyramidal indentation, all directional stress, including the longitudinal, transverse and shear stresses, around the hemispherical indentation were not concentrated around the indent edge. However, lots of line-style concentration zones, normal to the tensile loading direction, of the longitudinal stress, $\sigma_{yy}$, for both indents were discontinuously and locally scattered over the testing section, including the indent.

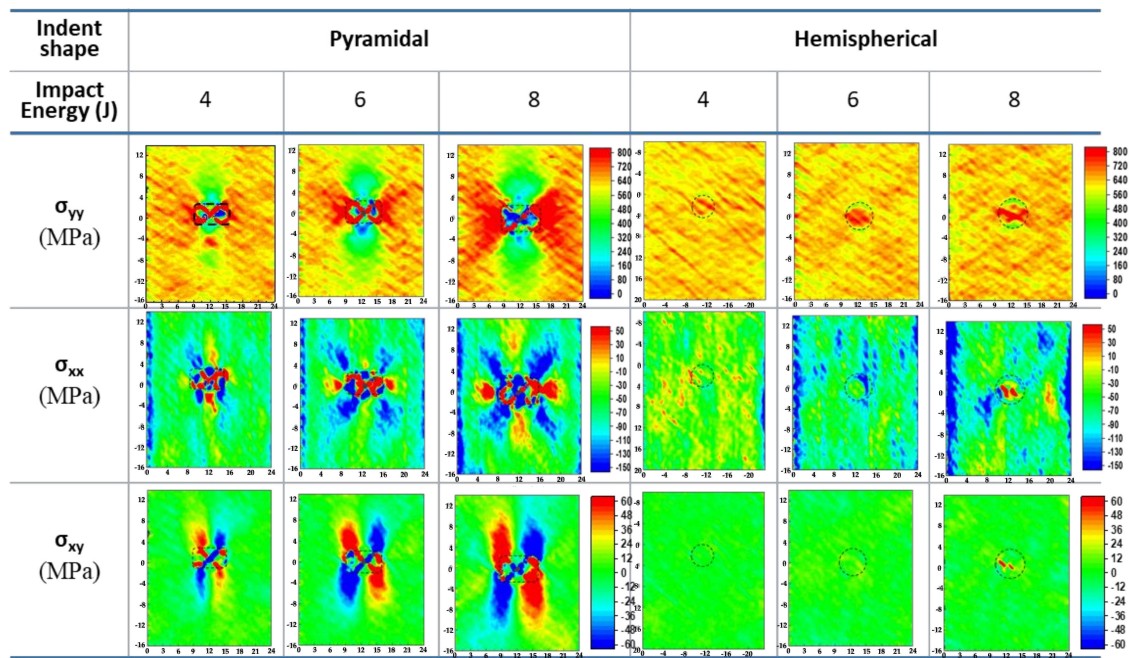

**Figure 9.** Typical directional stress, longitudinal ($\sigma_{yy}$), transverse ($\sigma_{xx}$) and shear ($\sigma_{xy}$) stresses, distribution in the vicinity of the indents for TRI GFRP laminates (at tensile stress of 600 MPa).

From the full-field stress distribution obtained from DIC measurement, as shown in Figures 9 and 10, stress distribution ahead of the indentation can be quantitatively compared considering impact conditions, such as impact energy and impactor head shape. In Figure 11, the normalized longitudinal stresses, which are the ratios of the longitudinal stresses to the nominal stress applied to the laminate specimen, ahead of the indent were presented. For the impacted TRI GFRP laminate, as can be found in Figure 11a, the maximum ratios ahead of the pyramidal indent were considerably higher than those ahead of the hemispherical indent, and the ratios were dependent on the impact energy levels. Here, those at the pyramidal indent boundary were 3.3, 2.6 and 1.6 at the impact energy levels of 8 J, 6 J, and 4 J, respectively, while the ratio at the hemispherical indent boundary was in the range of 1.3 to 1.1 Thus, it is found that the longitudinal stress concentration became higher as the impact energy level increased, and the pyramidal impactor had a greater influence on the stress concentration than the hemispherical impactor. In contrast, for the UD GFRP laminate, the maximum ratio at the impact energy of 8 J approached to 1.2, irrespective of the indent shapes, and the ratios for both indent shapes were less than 1.1 at the impact energy level of 6 and 4 J. Therefore, it can be said that, for the impacted UD GFRP laminate, the stress concentration at the impact energy level greater than 6 J was slightly high, but the concentration at all energy levels was relatively insignificant.

Dependency of the failure tensile strength of the impacted laminate on the indent shape and impact energy level is shown in Figure 8. For the TRI GFRP laminate impacted with the pyramidal impactor, the failure tensile strength was considerably reduced with increasing impact energy level, which corresponded considerably well to a tendency for the stress concentration around the indent to be higher with increasing impact energy. Moreover, difference in failure tensile strength reductions between the hemispherical and pyramidal indents also corresponded quite well to relative difference in stress concentrations between these two indents. Here, the strengths for the pyramidal and hemispherical indents were reduced by 13–18% and 3–13%, respectively, as can be found from Figure 8. Therefore, it is found that the reduction in the residual strength of the TRI GFRP laminate with increase of impact energy level could be associated with the high stress concentration depending on the impact energy. In comparison, for the UD GFRP laminate, both hemispherical and pyramidal indentations caused an insignificant stress concentration around the indent boundary, less than 1.1, as presented in Figure 11b, and also did not lead to a significant failure strength reduction, which was less than 4%, as can be

found in Figure 8. Therefore, it is found that the impact indents in the UD GFRP laminate generated insignificant stress concentration around the indent, irrespective of indent shape, and also did not lead to significant strength reduction associated with the stress concentration. This may be caused by a relatively low damage generation and progress initiated from the indent in UD laminate, which can be found in Figure 6.

**Figure 10.** Typical directional stress, longitudinal ($\sigma_{yy}$), transverse ($\sigma_{xx}$) and shear ($\sigma_{xy}$) stresses, distribution in the vicinity of the indents for UD GFRP laminates (at tensile stress of 700 MPa).

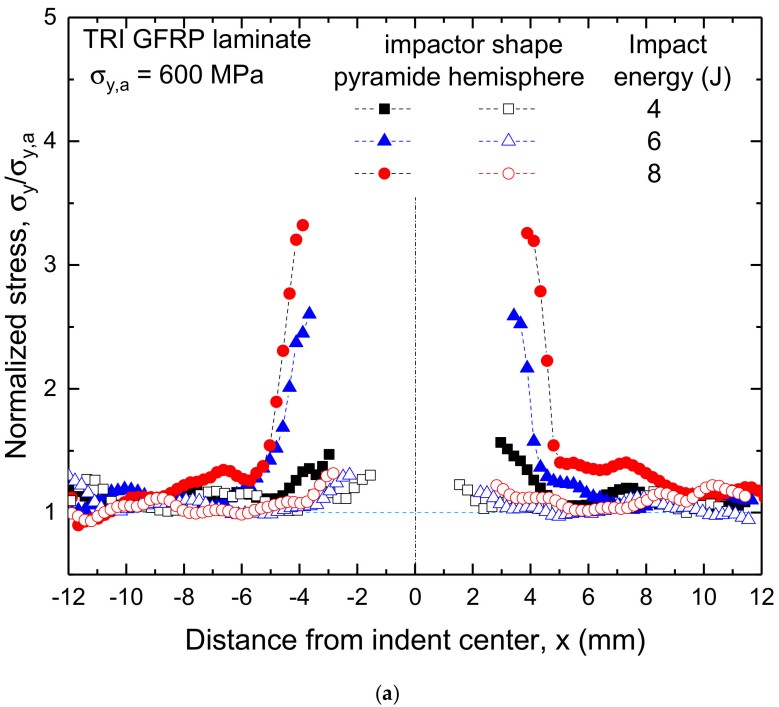

(**a**)

**Figure 11.** *Cont.*

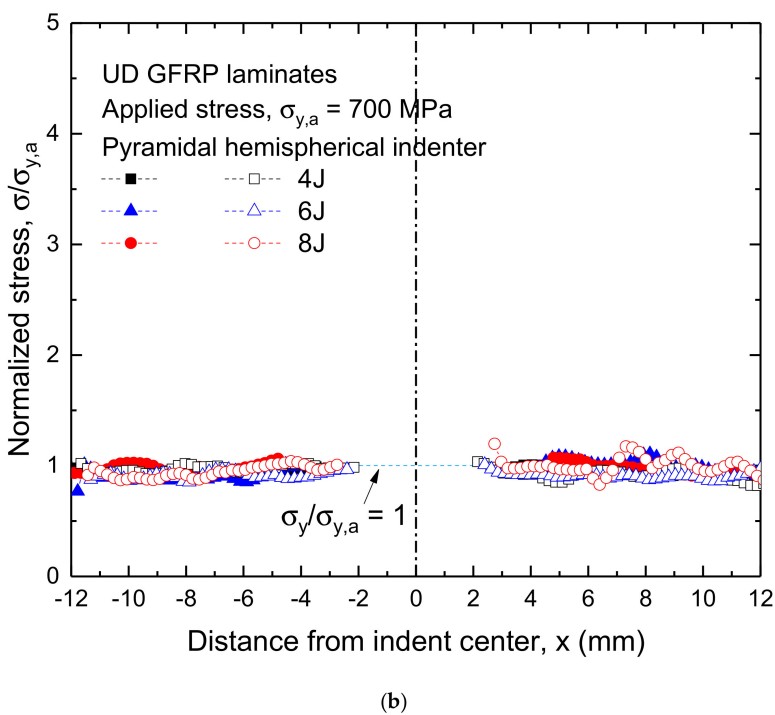

(**b**)

**Figure 11.** Normalized longitudinal stresses ahead of the indent for the impacted (**a**) TRI and (**b**) UD GFRP laminate.

Therefore, from this DIC measurement, a surface stress distribution around the impact indent could be determined, and the local stress concentration was found to be closely associated to the reduction of residual tensile strength of the impacted laminates. In particular, the local stress concentration severity, depending on the impact energy and indent geometry, around the indent could be quantitatively examined. Therefore, it may be useful to understand the fracture strength assessment of the GFRP laminates after impact. This quantitative mechanical stress approach on the impact damage in the composite laminates may be progressed with 3D modelling and analysis of the damage.

## 4. Summary and Conclusions

The impact response for the TRI and UD GFRP laminates after impact, including residual failure strength reduction, damage patterns and stress concentration, was investigated, considering the effects of the impact energy, impactor shapes and laminate structure. Furthermore, impact damage patterns were investigated and stress concentration around the impact indents boundary was evaluated from the full-field stress distribution, determined by DIC measurement, over the impacted area.

(1) The TRI GFRP laminate was remarkably more susceptible to the impact indentation than the UD laminate. Reduction of residual in-plane tensile strength for the impacted TRI laminate was relatively more severe than that for the impacted UD laminate.

(2) Reduction of residual in-plane tensile strength for the TRI and UD GFRP laminates was dependent on the impact energy level and indent shape. The strength reduction became greater with increasing impact energy level, and the pyramidal indent had a relatively greater influence on the residual strength reduction than the hemispherical indent.

(3) Full-field directional stress, including longitudinal ($\sigma_{yy}$), transverse ($\sigma_{xx}$) and shear ($\sigma_{xy}$) stress, distributions in the vicinity of the pyramidal and hemispherical indents for the TRI and UD GFRP laminates were determined from the DIC measurement using a linear elastic orthotropic relationship. These stresses were relatively significantly concentrated in the vicinity of the pyramidal indent for the TRI laminate, compared to those for UD laminate. Stress concentrations in the vicinity of the pyramidal and hemispherical indents for the TRI laminate were relatively

higher than those for the UD laminate. Furthermore, the stress concentration became higher with increasing impact energy levels.

(4)  The residual strength reduction of the impacted GFRP laminates may be considerably closely associated with the stress concentration in the vicinity of the indent.

(5)  The impact damage patterns created in the TRI and UD GFRP laminates were dependent on the impactor head shape, but the dominant patterns were matrix cracking, delamination, debonding and fiber breakage.

**Author Contributions:** Conceptualization, J.-I.K. and Y.-H.H.; methodology, J.-I.K. and Y.-H.H.; formal analysis, J.-I.K.; investigation, J.-I.K. and Y.-H.H.; data curation, J.-I.K.; writing—Original draft preparation, J.-I.K.; writing—Review and editing, Y.-H.H. and Y.-H.K.; supervision, Y.-H.H. and Y.-H.K.; funding acquisition, Y.-H.H. All authors have read and agreed to the published version of the manuscript.

**Funding:** This research received no external funding.

**Acknowledgments:** This work was supported by the KETEP (Korea Institute of Energy Technology Evaluation and Planning) granted by the Korea government (MSIT No. 20153010024470) and KRISS.

**Conflicts of Interest:** The authors declare no conflict of interest.

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
