# Peer review of "Static Residual Tensile Strength Response of GFRP Composite Laminates Subjected to Low-Velocity Impact"

_applsci, doi:10.3390/app10165480_

Round 1

Reviewer 1 Report

The article investigates the residual tensile strength after an impact event of GFRP. Influence of impactor head shape as well as impact energy on the damage patterns of two different composite lay-ups is analysed.  This is an interesting topic, but was already discussed in previous publications. Therefore, the novelty is somehow limited and a more detailed introduction would be helpful to put the work at hand into the appropriate context.

As most of the research on impact damage in composites focuses on residual compressive strength as the critical parameter, the investigation of tensile strength and the focus on the impactor shape is an interesting approach. Internal damage and stress distributions around the impact are analysed and the results are well presented. The paper is thus an interesting contribution to current research in this field and rated for publication after a revision.

For detailled comments, please refer to the attached file.

Author Response

The authors appreciate the reviewer’s thorough and kind comments on the manuscript entitled by “Static residual tensile strength response of GFRP composite laminates subjected to low-velocity impact”. The authors revised the manuscript carefully according to the reviewer’s comment.

The authors prepared replies to the reviewer's commnets carefully and please refer to the attached file

Reviewer 2 Report

This is a good paper, providing interesting results and valuable information. However, to be published it requires to be enriched with a better introduction, and, most importantly, with a greater level of details in the methodology and materials used; because, as it is now, the results are not independently reproducible.

I suggest a thorough review of the language, grammar, and spelling, maybe by a native English speaker expert in composites, as some sentences are incorrect and there are a few “slip of the pen”…

Abstract

In the first line you repeat “impact” 4 times in two lines… please rephrase to remove repetitions.

I’d suggest removing the acronyms from the abstract.

Introduction

The introduction needs to be further improved to be wider and more in depth. Considering the contiguity of topics and that the adopted methodology is substantially the same I’d suggest to quote the paper from Prato et al. “Post-Impact Behaviour of Pseudo-Ductile Thin-Ply Angle-Ply Hybrid Composites” ( https://doi.org/10.3390/ma12040579 ).

Experimental programme

The information provided regarding the material choice are insufficient. You need to be specific in terms of material (makers and types), format (prepreg, dry fibre), manufacturing methods (autoclave curing, infusion) and provide basic information, maybe with a table, regarding the laminate properties (stiffness, strength, fibre volume fraction).

“the machine had crosshead mass of 78 N” a mass measured in N? are you sure?

Many information are missing from your experimental set-up description, for example: how are the specimens supported while subjected to impact? Remember that another researched should be able to replicate your work by only reading your paper.

Why not providing a drawing of impactors and specimen…

The description of the tensile test set-up is insufficient.

The DIC testing set-up is quite good, however spatial and strain resolutions are missing.

Results and discussion

“which may be from 101 presence of fibers break which would be caused by a locally dynamic impact process” not clear what this means, please explain the “locally dynamic impact”.

The information between line 108 and 119 could be more effectively provided in a well organised table, this would also allow to provide information about all the investigated loading cases.

In the “Reduction of residual static strength” section you really need to provide the information regarding the baseline tensile tests done on the undamaged specimens. I would suggest again to summarise the information provided in Fig. 7 in a table as well.

Summary and Conclusions

I think that conclusion (1) is unclear, please rephrase.

You need to provide a closing paragraph to “locate” your paper in the wider scientific context, e.g. explaining why the results you obtained are significant, both on a scientifical and industrial point of view, and suggesting how to take them forward, e.g. further experimental work and/or modelling activities.

My final note is that this paper is very good, the results are well presented and conclusions are well supported by them. However, it is very weak in terms of materials and methodology description, and needs to be better contextualised considering the current available literature and the relevance of the results to academia and industry.

Author Response

The authors appreciate the reviewer’s thorough and kind comments on the manuscript entitled by “Static residual tensile strength response of GFRP composite laminates subjected to low-velocity impact”. The authors revised the manuscript carefully according to the reviewer’s comment.

The authors prepared replies tot he reviewer's comments carefully, and please refer to the attached file.

Round 2

Reviewer 2 Report

The paper is now suitable for publication